# The Role of AI in Warehouse Digital Twins: Literature Review †

**Adnane Drissi Elbouzidi** [1,2,*] **, Abdessamad Ait El Cadi** [3,4] **, Robert Pellerin** [5] **, Samir Lamouri** [1] **, Estefania Tobon Valencia** [2] **and Marie-Jane Bélanger** [5]

[1] LAMIH, Arts et Métiers ParisTech, 51 Bd de l'Hôpital, 75013 Paris, France; samir.lamouri@ensam.eu
[2] Groupe Square Management cabinet Flow&Co., Square Research Center, 173 Avenue Achille Peretti, 92200 Neuilly-sur-Seine, France; e.tobonvalencia@flowandco.fr
[3] LAMIH, CNRS, UMR 8201, Université Polytechnique Hauts-de-France, 59313 Valenciennes, France; abdessamad.aitelcadi@uphf.fr
[4] INSA Hauts-de-France, Université Polytechnique Hauts-de-France, 59313 Valenciennes, France
[5] Polytechnique Montreal, 2500 Chemin de Polytechnique, Montréal, QC H3T 1J4, Canada; robert.pellerin@polymtl.ca (R.P.); marie-jane.belanger@productique.quebec (M.-J.B.)
* Correspondence: adnane.drissi_elbouzidi@ensam.eu
† This paper is an extended version of the paper "The Role of AI in Warehouse Digital Twins" published in 34th European Modeling & Simulation Symposium (EMSS), Rome, Italy, 19–21 September 2022.

**Abstract:** In the era of industry 5.0, digital twins (DTs) play an increasingly pivotal role in contemporary society. Despite the literature's lack of a consistent definition, DTs have been applied to numerous areas as virtual replicas of physical objects, machines, or systems, particularly in manufacturing, production, and operations. One of the major advantages of digital twins is their ability to supervise the system's evolution and run simulations, making them connected and capable of supporting decision-making. Additionally, they are highly compatible with artificial intelligence (AI) as they can be mapped to all data types and intelligence associated with the physical system. Given their potential benefits, it is surprising that the utilization of DTs for warehouse management has been relatively neglected over the years, despite its importance in ensuring supply chain and production uptime. Effective warehouse management is crucial for ensuring supply chain and production continuity in both manufacturing and retail operations. It also involves uncertain material handling operations, making it challenging to control the activity. This paper aims to evaluate the synergies between AI and digital twins as state-of-the-art technologies and examines warehouse digital twins' (WDT) use cases to assess the maturity of AI applications within WDT, including techniques, objectives, and challenges. We also identify inconsistencies and research gaps, which pave the way for future development and innovation. Ultimately, this research work's findings can contribute to improving warehouse management, supply chain optimization, and operational efficiency in various industries.

**Keywords:** digital twins; warehouse; material handling; artificial intelligence; machine learning





## 1. Introduction

Intralogistics is undoubtedly a crucial component of both manufacturing efficiency and customer satisfaction. Material handling, in particular, can account for 15 to 70% of production costs, underscoring the need for optimized warehouse operations [1]. Additionally, order preparation costs are believed to make up as much as 55% of the overall expenses incurred in a warehouse [2]. Material handling is also one of the most hazardous industrial processes, accounting for up to 50% of all industrial injuries. Given the evolving market landscape and increasing customer demands, companies have had to adapt their strategies to meet these shifting requirements while remaining financially viable. A recent survey by LaserShip revealed that more than 60% of consumers are willing to pay additional fees for same-day delivery, highlighting the importance of streamlined and transparent warehouse processes [3]. Customers now expect product personalization and ethical and

environmental responsibility from their suppliers, further complicating warehouse management. As a result, it is critical to implement a dynamic, straightforward, and fully transparent process to tackle the myriad of uncertainties that plague warehouse management, including changes in supply chain structures, demand seasonality, and fluctuations in transportation costs [4].

Contemporary society is currently experiencing a digital and ethical revolution [5]. Having already benefited from industry 4.0 technologies, companies and society are shifting more profoundly towards virtual reality, in which the physical and digital worlds are increasingly intertwined, known as industry 5.0 [6]. With technology as the enabling tool and societal needs as the goal, industry 5.0 aims to promote sustainability, social responsibility, and resilience by integrating digital technologies.

Digital twin (DT) is one of the industry 4.0 pillars that has rapidly gained traction over the past decade and it shows significant promise for enhancing warehouse management [7]. However, its precise definition has been a topic of ongoing debate in both academia and industry. The concept originated from simulation, an ever-evolving modeling approach dating back to 1960 [8]. The benefits of simulation in relation to industry 4.0 and manufacturing have proven to be numerous and varied. It can be used to develop a process of analysis and optimization of scenarios proposed in a virtual environment, allowing for the reduction of risk and the validation of processes before implementation [9]. Through real-time simulation, a continuous optimization of resources can be guaranteed, and a shared platform with suppliers for order allocation and management can considerably reduce the complexity of relations with suppliers and customers, leading to a reduction in lead times and internal handling. Therefore, simulation can be a valuable tool in optimizing logistics in the industry 4.0 context. Stemming from simulation as their core, DTs are expected to provide a fully connected and continuously evolving virtual replica of their physical counterparts, enabled by IoT technology that facilitates data analytics and simulation [10]. Moreover, DTs possess smart capabilities that will allow them to not only monitor changes in the warehouse but also promote them to enhance performance.

Furthermore, digitalization in warehousing is not novel, yet it has garnered more attention in recent years, driven by industry 4.0. In the 1990s, the early adoption of Enterprise Resource Planning (ERP) systems in logistics and supply chain management initiated efforts to improve data availability in material handling [11]. As the journey towards digitalization progressed, the implementation of Warehouse Management Systems (WMS), Internet of Things (IoT), automation, and big data have played significant roles in streamlining and standardizing intralogistics operations. Consequently, intralogistics and warehouses now possess a wealth of data that is often underutilized by decision-makers but could serve as a solid foundation for implementing innovative AI applications. However, large-scale automation has only been achievable for a few prominent companies, such as Amazon and Google [12]. A survey conducted in 2022 revealed that 79% of warehouses use some warehouse management systems (WMS), and 16% have adopted a "goods-to-person" warehousing solution [13]. Nevertheless, the data in warehouses remains largely untamed and disorganized. Many companies worldwide use warehouse management systems inefficiently, collecting data but often leaving it unattended, untreated, and unused. AI-based decision-making methods offer the potential to further enhance the features of DTs by enabling them to analyze and interpret the data collected by IoT devices autonomously. AI algorithms can identify patterns in data that can be exploited to predict future values of a variable, especially when the variables are complex or impossible to calculate. AI has found many applications in the logistics field, including inventory optimization and production planning [14,15]. In this way, advanced data analytics can be leveraged to transform DTs from mere data repositories into a source of knowledge or wisdom, driving optimized warehouse management.

AI and DT are heavily reliant on data, and in recent years, warehouses and supply chains have emerged as significant data sources and data-driven processes. The advent of AI-assisted DT presents an opportunity to address this problem. By structur-

ing data and serving as a unified source of truth and information, DTs can offer global oversight of warehousing activities, allowing for informed decision-making through data-driven insights. Consequently, implementing DT technology can help businesses transform chaotic and unwieldy data in their warehouses into valuable information that can drive improved performance.

Having established the complementarity between AI and DTs in warehouse management, this paper aims to provide an in-depth analysis of the state-of-the-art by assessing the maturity of AI applications within the DT paradigm. Through a comprehensive review of the existing literature, we seek to address the following research questions:

- What AI techniques are mostly used for warehouse management under the DT paradigm?
- How is AI employed to ensure and elevate WDT functions?
- What are the challenges and barriers to adopting WDT and AI in warehouses?

By addressing these questions, this paper seeks to contribute to a better understanding of the potential and limitations of AI and DTs in warehouse management and to provide insights into how these technologies can be effectively integrated to drive operational excellence and competitive advantage in the digital age.

The rest of this paper is structured as follows. Section 2 describes the methodology used for the systematic literature review and highlights the contribution of this study compared to existing ones. A bibliometric analysis is then presented to evaluate the keywords used for the literature review in Section 3. Section 4 introduces the analysis framework applied in this study, explaining the criteria and methods employed for selecting and evaluating the articles. Section 5 presents and discusses the findings of the review. Finally, Section 6 concludes the paper with a discussion of the overall challenges and perspectives, highlighting the potential of AI and DTs to revolutionize warehouse management and offering recommendations for future research.

## 2. Research Methodology

The present study employed a systematic literature review method proposed by [16], which has been successfully used by other researchers to gain insights from the scientific literature. The rigorous systematic review methodology allowed for a detailed analysis of each chosen article. The primary objective of this literature review was to showcase the potential of WDT as a novel decision-making tool for intralogistics. This study is domain-specific and should not be considered a replacement for generalized contributions regarding digital twins. Instead, the analysis builds upon the knowledge derived from several literature reviews that laid the foundation for digital twin research from a general perspective, such as [17–19].

While intralogistics is part of the supply chain, the scope of this review did not include the latter due to the risk of straying from warehousing activities as a production process rather than a network. Instead, this state-of-the-art review focuses on papers covering AI and DT technologies to optimize in-store warehouse activities. Since AI is a vast research field comprising numerous algorithms, detailed AI algorithms described by [14] were incorporated into the research query, along with general AI subgroups such as machine learning and deep learning, to encompass a broader range of AI-induced DTs.

To ensure a relevant literature inspection, specific keywords were targeted in titles, abstracts, and keywords, including ("Digital Twin" OR "Digital Twins") AND ("Warehouse" OR Warehousing" OR "Material Handling" OR "Inventory" OR "Packing" OR "Store" OR "Storage") AND ("Deep Learning" OR "Artificial Intelligence" OR "Machine Learning" OR "AI" OR "ML" OR "Neural Networks" OR "Regression" OR "Clustering" OR "Sarsa" OR "Nearest Neighbors" OR "Q-learning" OR "Decision Tree").

The query was executed between 5 April 2022 and 22 March 2023 in two scientific databases, ScienceDirect and SCOPUS, which yielded 277 articles. Only publications labeled as "Research Articles" in ScienceDirect and "Conference paper" or "Article" in SCOPUS were included initially to capture articles presenting application models. After filtering out duplicates, a review of titles and abstracts allowed for excluding articles

unrelated to AI and DTs. Further, a full-text analysis was performed, resulting in a final selection of papers that fit the research questions and two additional papers selected from the reviewed of the shortlist. The sample size obtained comprises 22 scientific papers. The selected references either display a fully embedded AI and digital twin application or explain the possible relationships between both technologies and how one could exploit the other. The article selection methodology is described in Figure 1.

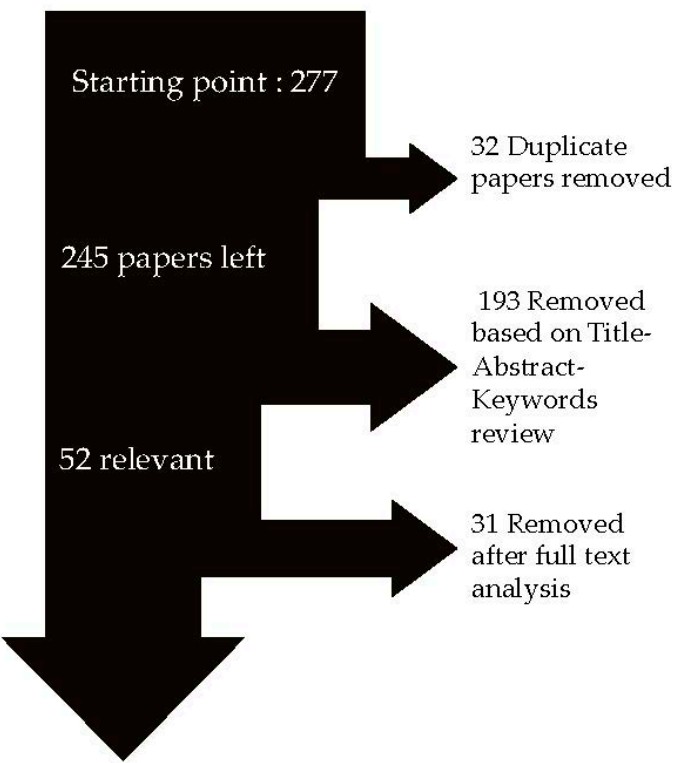

**Figure 1.** The search strategy used to capture the scientific literature.

### 3. Bibliometric Analysis

Digital twin as a concept is tightly knit with all of industry 4.0 technology, primarily simulation, machine learning, hyperphysical systems, IoT, etc. Some research questions the novelty of digital twins, given how it is very comparable with simulation [20]. To counter that, ref. [8] presents an evolution of simulation to digital twins, showcasing how the two concepts could be distinguished. The authors mention that the first simulations, dating back to 1960, were for applications limited to very specific topics such as mechanical parts. Around 1985, computer simulation tools appeared to support the conception and design of parts. Simulation-based system design dates from the 2000s with the introduction of complementary disciplines such as model-based systems engineering. Simulation has enabled a systems approach for multi-level and disciplinary systems with a wide range of applications.

Digital twins are often considered the second coming of simulation as they are built from the construction of a virtual environment emulating the physical system and constantly interacting with it. This is why DTs often require IoT to collect and transfer data to the virtual replica and then realize actions in the physical space. Given the duality of the technology from both the physical and cyber components, it is easily mistaken for cyber-physical systems.

There seems to be no clear boundary between these terms, so a bibliometric analysis was performed to assess the chosen keywords. The study was done using VOSviewer 1.6.19, a software the University of Leiden developed to draw insights from the scientific literature. The bibliometric analysis followed a simplified methodology of that used to choose the final article sample (cf. Figure 1). At first, a general search using only "digital twin" was targeted in titles, abstracts, and keywords. The query was performed on 10 January 2023 during which no title nor abstract review was done as this could introduce a bias into the results due to the authors' influence. The bibliometric analysis focused on the keywords defined by the authors for all the papers resulting from the search query. The objective was to visualize and identify concepts related to DTs that might have needed to be included in the main literature review query.

Secondly, two research queries were introduced to Scopus:

- "Digital Twin" AND ("warehouse" OR "warehousing" OR "material handling" OR "inventory" OR "packing" OR "store" OR "storage"),
- "cyber" AND" physical" and "system" AND ("warehouse" OR "warehousing" OR "material handling" OR "inventory" OR "packing" OR "store" OR "storage").

To represent the results, the network visualization from VOSviewer was employed. In such a network, the nodes represent the keywords, their sizes reflect the keyword importance determined by the number of occurrences, and the links between the nodes represent their co-occurrence. Furthermore, the relatedness between two terms is represented through their spatial distance in the network: two keywords closely related will be spatially closer. The networks are presented in Figures 2–4.

The first network represents all concepts in the scientific literature related to DTs alone (Figure 2). The number of occurrences for the showcased items has been set to more than 100 times. The network is made of three primary cohesive clusters related to artificial intelligence in yellow; industry 4.0 and CPS in red, and simulation in green. Results from the bibliometric study suggest that DT as a concept is related to simulation and cyber-physical systems, with a stronger link towards simulation.

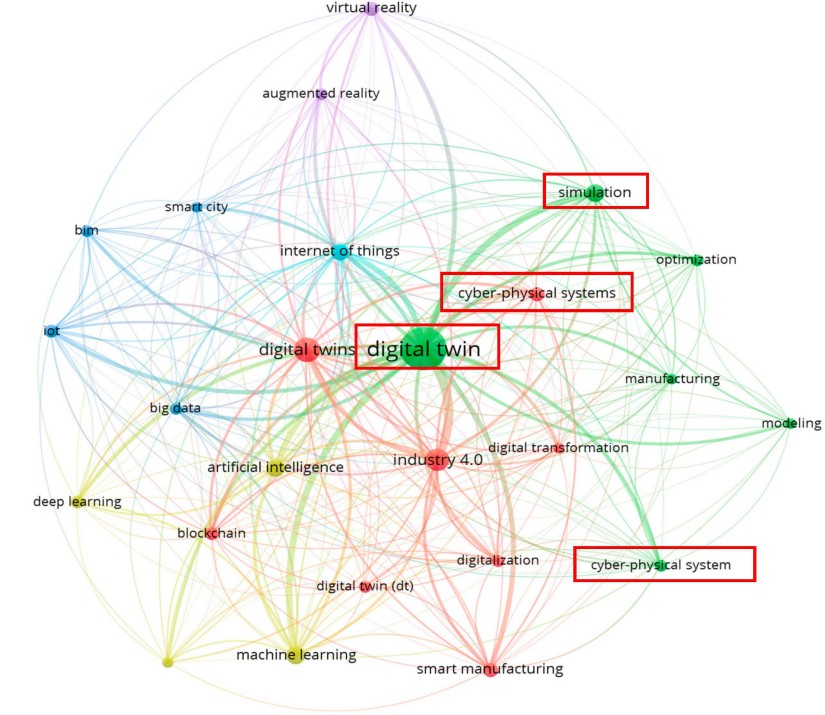

**Figure 2.** Network visualization for "Digital Twin" alone.

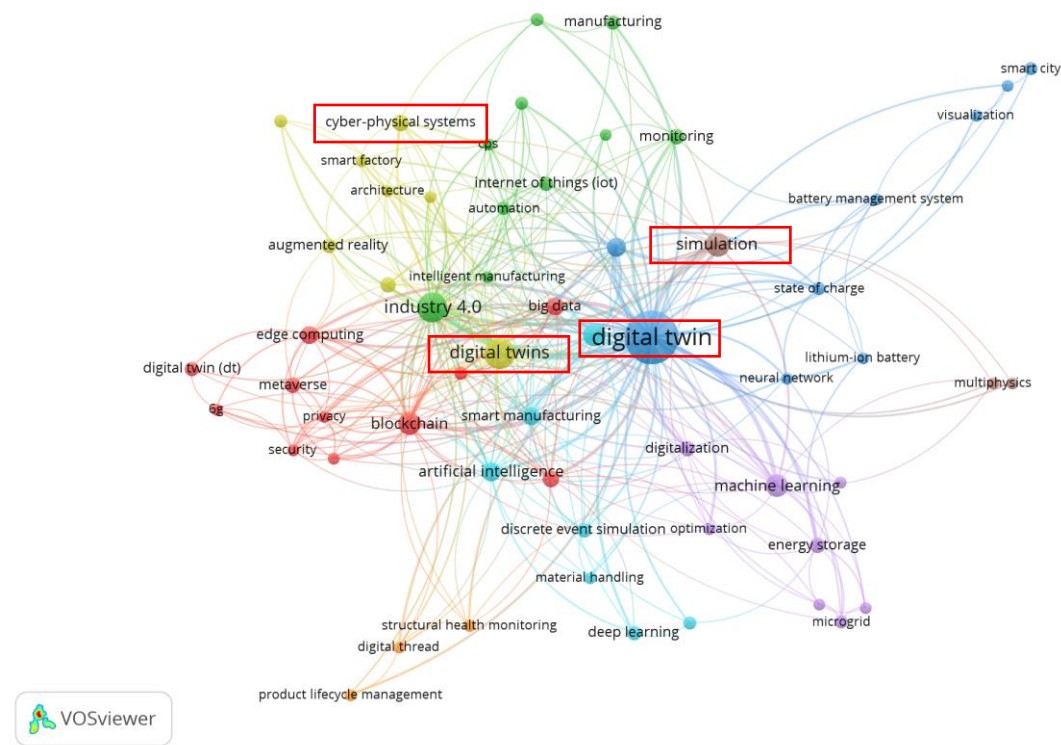

**Figure 3.** Network visualization for "Digital Twin" in the context of intralogistics.

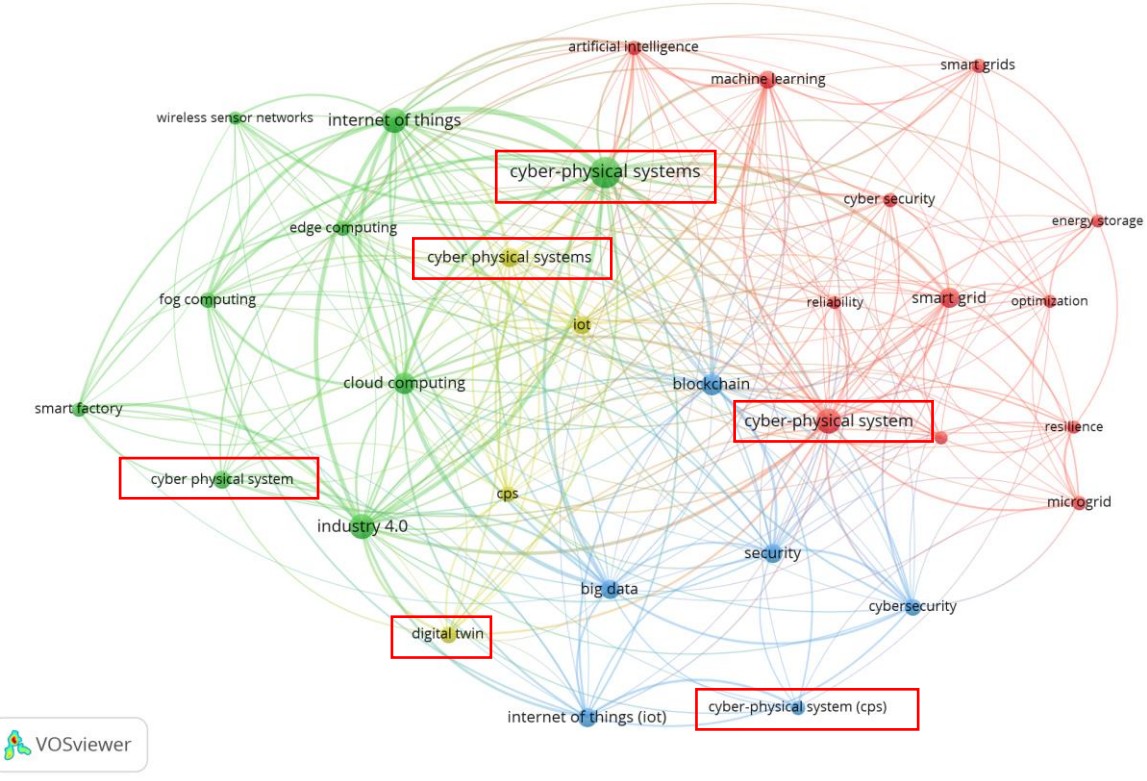

**Figure 4.** Network visualization for CPS in the context of warehousing.

Results from the bibliometric study suggest that "Simulation" may not be a common keyword to find in cyber-physical systems in warehousing research. It does not appear, probably due to the filter excluding keywords with number of occurrences below 20, the sample size being 1883.



Analyzing the relatedness between "DTs", "cyber-physical systems", and "simulation" by their spatial distance on the networks provides an idea of how these concepts are associated: they are spatially closer on the general network (Figure 2) than on the "Warehousing" networks (Figures 3 and 4). This suggests that DTs tend to relate more often to simulation than cyber-physical systems, which have weak links to DTs in all networks and are much farther from DTs in the query focused on CPS. Such a relation aligns with the statement in [21] which indicates that in manufacturing, both CPS and DTs include the physical and the cyber/digital parts The essence of CPS, however, is to add new capabilities to physical systems using computation and communication, which intensively interact with the physical processes. Compared to DTs, CPS more strongly emphasizes the cyber world's powerful computing and communication capabilities, which can enhance the accuracy and efficiency of the physical world. Furthermore, all the proposed CPS architectures focus on control through controllers and sensors rather than on mirrored models. As for simulation, the concept is integral to digital twins, as detailed further in this review. The emergence of DTs relies heavily on simulation capabilities and could be considered its starting point [18,20]. However, simulation models tend to be static in time as they focus mainly on the virtual side of digital twins. In contrast, DTs require data transmission between the physical and virtual world to be fully realized [18].

From the bibliometric analysis, it could be concluded that using only "digital twin" in the query keywords is appropriate enough, as this allows for identifying a large sample of recent papers, enabling the identification of new trends. Finally, even if "simulation" is closely related to "Digital Twins", it covers a vast domain that can deviate from the focus of this review and lacks "twinning" requirements.

## 4. Analysis Framework

This paper touches on the integration of AI in digital twins. The proposed analysis axes seek to challenge the scientific literature on digital twins applied to optimize intralogistics and warehousing activities. The objective is to differentiate what qualifies as a DT and how it is used in collaboration with AI.

Considering the severe need of both AI and DTs for data and its scarcity in warehousing and logistics, special care was given to determining data requirements for full implementation of the technology.

This section presents the main axes that build the analytical framework which will be employed to harness insights from the final sample of 22 scientific articles.

### 4.1. Digital Twin

DTs are comprehensive digital representations of physical assets, comprising their design, configuration, state, and behavior [17]. A more grounded interpretation of the concept considers DT as the effortless data integration between a physical and virtual machine in either direction [18]. Instead of focusing on the definition of DT, ref. [18] made the distinction between what it is not and why, which led to the identification of the following levels of DTs:

1. Digital Model: There is no automatic data exchange between the physical and digital worlds. Once the model is created, a change made to the physical object has no impact on it.
2. Digital Shadow: A digital shadow is a digital model with a one-way data flow from the physical to the digital objects. A change in the state of the physical object leads to a change in the digital representation.
3. Digital Twin: the data flow between the two counterparts is bidirectional. A change to the physical object automatically changes its virtual replica and vice versa.

To further distinguish the maturity levels of the papers under study, an analysis of the characteristics of DTs detailed in the existing literature put the shortlisted article to the test. Ref. [19] described the characteristics of cognitive digital twins as being DT-based, cognition, full lifecycle management, autonomy capability, and continuous evolving.

Alternatively, ref. [17] described digital twins as autonomous, context-aware, and adaptive virtual replicas. Considering the works mentioned above, relevant DT characteristics were identified for this study, reflecting a combination of the two visions above. Additionally, AI's potential integration and use in fulfilling these characteristics were investigated. The definition of each characteristic is provided below:

- Context-awareness (CA) is the ability to distinguish incoming stimuli meaningfully. It encompasses more than just IoT and information systems (IS), extending to representing diverse situations in a virtual copy.
- Autonomy (Auto) is the DT's ability to function independently without human intervention. This capability empowers the system to take action and make decisions based on pre-determined rules or learned behaviors, streamlining the decision-making process without human assistance or a minimum level of human intervention.
- Continuous evolving (CE) is the ability of a DT system to grow and evolve with the real system throughout its lifecycle. DT systems should continuously update themselves based on changing data, information, and knowledge from the real system and all other interconnected software. This feature allows the DT system to adapt to new environmental conditions and changes, ensuring that it remains relevant and effective over time.
- Full lifecycle management (FLM) allows the model to cover different phases across the entire system lifecycle. FLM includes the beginning of life (BOL), such as design, building, and testing; the middle of life (MOL), such as operating, usage, and maintenance; and the end-of-life (EOL), such as disassembly, recycling, and remanufacturing. By addressing all lifecycle phases, FLM enables the DT system to be more sustainable, efficient, and effective over the long term.

The characteristics of DT discussed above are considered essential requirements for a twinning paradigm, enabling higher levels of accuracy and a more realistic representation of the system. The specific levels and characteristics of DT may vary depending on the model's objectives and its intended application. Nonetheless, this literature review aims to assess how far these characteristics can be established and their interplay with data and artificial intelligence (AI).

### 4.2. Artificial Intelligence

AI has existed since the Dartmouth Summer Research Project on Artificial Intelligence in 1956, but its evolution has been slow and rocky. Even today, there is yet to be a clear consensus on what can be considered AI [22]. AI methods include analytic hierarchy process (AHP), fuzzy logic (FL), genetic algorithms, neural network (NN), and simulated annealing (SA).

Modern subsections of AI include machine learning (ML) and deep learning (DL), which are some of the most popular ones [23]. Although the terms are often used interchangeably, DL specifically refers to deep artificial neural networks and sometimes deep reinforcement learning, primarily ML techniques.

AI developers today have realized that training a system by demonstrating examples of desired input-output behavior is a much simpler and more efficient approach than manually programming the desired response for every conceivable input [23]. Adopting machine learning and deep learning techniques has paved the way for such training. These methods enable systems to learn from large volumes of data and gradually enhance their performance over time. However, the various AI techniques have different strengths and limitations, and the choice of which one to use depends on the specific application and the available data. For example, neural networks are good at recognizing patterns and making predictions based on past data, while genetic algorithms are more suited to optimization problems.

An examination of the various types of learning employed is deemed essential to garner a deeper understanding of the ML techniques utilized in the field. This will not only aid in summarizing the information surrounding these techniques but also facilitate

the identification of prevailing trends and research perspectives. Drawing on the seminal work of [23], three main types of machine learning can be distinguished below. The main discrepancy between these types lies in whether the training dataset is labeled or not.

- Supervised learning (SL): the algorithm is provided with a clearly defined set of input features X and corresponding output labels Y. Supervised learning can be used in intralogistics to predict demand for specific products, to optimize inventory levels, or to predict delivery times.
- Unsupervised learning (UL): the algorithm is provided only input features X. The goal is to find patterns or structures within the data that can be used to group similar data points or to identify outliers using techniques such as cluster analysis. UL is typically used when there is no clear understanding of the underlying structure of the data or when there is no prior knowledge about the data. UL, such as clustering, can be used in intralogistics to identify similar groups of products or to cluster similar customers based on their buying behavior.
- Reinforcement learning (RL): involves an agent that learns by interacting with an environment and receiving rewards or punishments based on its actions. The learner aims to maximize the cumulative reward value over time through trial and error. Reinforcement learning is commonly used in tasks such as game playing, robotics, and autonomous navigation. RL can be applied to train an automated guided vehicle (AGV) in a warehouse to navigate through the facility while avoiding obstacles and maximizing the number of delivered packages. By interacting with the environment, the AGV learns which actions lead to the most desirable outcomes and adjusts its behavior accordingly, gradually improving its performance over time. This allows for a more flexible and adaptive approach to learning in general, which responds to new challenges and changing environments without explicit programming.

### 4.3. Data

This section provides an in-depth exploration of data's role in replicating warehouses in the virtual space. It is undeniable that the application of AI and DTs is heavily reliant on data. Almost every application of AI and DTs can be traced back to data as the starting point. Notably, the use of data in warehousing is diverse and varies depending on several factors, such as availability, source, and application. Three primary data types are collected in warehouses from different sources, including manufacturing information systems, IoT devices, and manual data entry [14]. These data sources include:

- Environmental data (ED): such as temperature, humidity, and light intensity, could be crucial in decision-making processes or to represent and supervise the physical process accurately. Depending on the type of goods stored in the warehouse, these data may provide valuable insights into the most suitable storage conditions.
- Product data (PD): which entails information on inventory levels and storage locations is another key data type. Technologies such as radio frequency identification (RFID) can monitor storage locations and quantities, linking this information to the warehouse management system (WMS) and DTs for effective replenishment and stockkeeping.
- Handler data (HD) is the third type of data collected in warehouses, providing crucial information on workers and equipment, including their real-time locations. This data may be collected from workers' handheld devices, allowing for location tracking, and measuring other physical variables. It can also refer to equipment and automation data such as conveyors and AGVs.

It is also essential to consider data sources to judge the level of DT interoperability and connectivity to the physical world. Data sources refer to the different systems and technologies used to collect and manage data within the warehouse. They may include WMS, IoT devices, and other digital technologies. When evaluating the DT interoperability and connectivity level, it is crucial to consider the compatibility of these data sources. For instance, if a virtual warehouse utilizes a WMS to manage inventory and orders, the DTs

used should be able to interact with the WMS seamlessly to facilitate real-time data sharing, if not being integrated into it.

### 4.4. Intralogistics

Intralogistics refers to the management and optimization of material and information flows and processes within a company's facilities [24]. It involves planning, implementing, and controlling internal logistics processes, including handling, storage, and transportation, as well as using information and communication technologies to improve efficiency and productivity. Intralogistics aims to ensure the continuous flow of goods and information within a company's operations, from reception until delivery, in the most effective and efficient manner possible.

Warehouse operations can be performed manually, or be fully automated, utilizing robots and mechanical tools that require minimal human intervention (e.g., conveyors, stacker crane ... ), or a hybrid of human and machine working together. Warehousing activities vary depending on the product, industrial sector, and warehouse type. The warehouse logistics framework proposed by [3] is also targeted in the shortlist to ensure a comprehensive understanding of the subject matter. This framework encompasses several key activities involved in in-house logistics, including product/order arrival, put-away and preparation for storage, storage, order picking, preparations for shipping (packaging, accumulation, sortation), and shipping. By adopting this structure, a thorough understanding of the material flow and logistics involved in warehousing operations is gained. The framework can be a useful guide for practitioners and researchers seeking to map out and optimize warehouse performance and efficiency.

### 5. Results

The literature reviewed highlights the diverse range of AI applications in WDTs. These uses span various processes performed within the warehouse and related activities in asset management and synthetic sensing [25–28]. Interestingly, the papers examined in this study demonstrate that a blend of manual and automatic stores could be optimized by implementing WDTs and AI, with very few applications focused on the optimization of reception, shipping, and preparation of goods/deliveries. As a concept, DT comes at the intersection of cloud computing, simulation, and IoT. Being part of industry 4.0 [6,7], DT modeling is seldom approach-based and focuses mostly on technological requirements and connectivity. This may be due to the assumption that DT, being closely tied to IoT technology, would have readily accessible model parameters and data [10], which is why it is worth noting that a highly automated, fully mechanical system is not necessarily required to make use of AI in the context of WDTs [29]. Table 1 and Figure 5 summarize the findings of this literature review and the topics discussed either in depth or lightly in each reference.

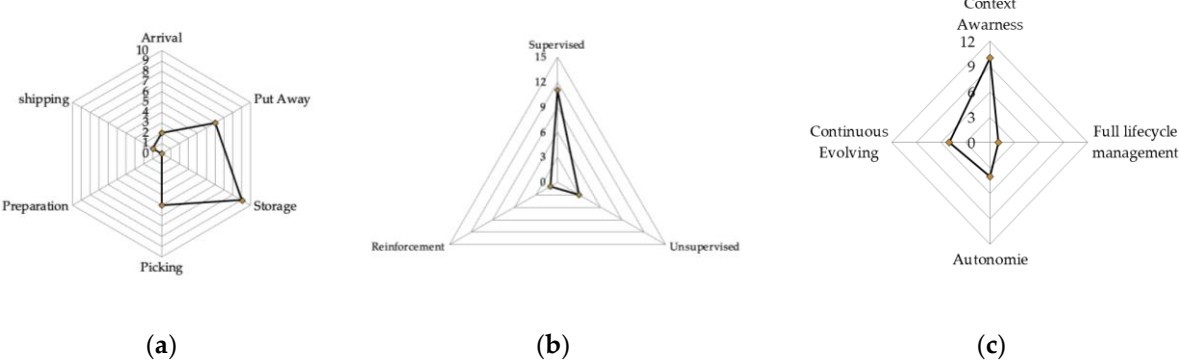

|  (**a**)  |  (**b**)  |  (**c**)  |

**Figure 5.** State of the art in WDT research regarding (**a**) intralogistics activities; (**b**) Machine learning and (**c**) Digital Twin characteristics.

### 5.1. What AI Technics Are Most Used for Warehouse Management under the DT Paradigm?

5.1.1. Artificial Intelligence

The literature reviewed in this study has shown a wide range of ML applications in WDTs, primarily focused on classification, and forecasting tasks. Neural networks (NNs), especially convolutional NNs, have been extensively used for classification tasks. For example, ref. [30] used a convolutional NN to monitor fruit freshness in stores by analyzing thermal images of bananas. In contrast, ref. [31] used a sparse autoencoder for abnormal stationary differentiation in cold storage warehouses. Object detection for inventory and asset inspection applications in buildings has been successfully accomplished using YOLOv2, as demonstrated by [26,32]. In addition, location estimation has been performed using deep learning techniques such as long short-term memory (LSTM) networks, as demonstrated by [33,34]. Ref. [25] utilized black hole optimization-based clustering to group the available supply demands based on time-frame-related objective function.

**Table 1.** Overview of the literature regarding digital twins and related concepts.

| | ML | | | Other AI Technics | | Level of DT | | | DT Characteristics | | | | Data Types | | | | Data Source | | | Warehouse Activities | | | | | |
| --- | --- | --- | --- | --- | --- | --- | --- | --- | --- | --- | --- | --- | --- | --- | --- | --- | --- | --- | --- | --- | --- | --- | --- | --- | --- |
| | SL | UL | RL | FL | GA | DM | DS | DT | CA | FLM | Auto | CE | Nature | ED | PD | HD | IS | IoT | Manuel Input | Arrival | Put Away | Storage | Picking | Preparation | Shipping |
| [30] | X | | | | | | | X | X | (X) | (X) | | R | X | | | X | X | X | | | X | | | |
| [20] | | | | | | | | X | | (X) | (X) | | | | | | | | | | | | | | |
| [35] | X | | | X | | | | X | X | | | | S | X | X | | X | X | | X | | X | X | | X |
| [36] | X | | | | | | X | | | | | X | R | X | | | | X | | | | | | | |
| [37] | | | | | | | X | | X | | (X) | | R | X | | | X | X | | | X | X | | | |
| [38] | | X | | | | | (X) | | X | | | | S | X | | | | | | | | X | | | |
| [31] | | X | | | | | | X | X | | (X) | X | R | X | | X | X | X | | | X | X | X | | |
| [28] | | | | | | | | X | | | | | | | | | | | | | | | | | |
| [27] | X | X | | | | | | X | | | | | | | | X | | | | | | | | | |
| [39] | | | | | | | | X | | | | | | | | | | | | | | | | | |
| [32] | X | | | | | X | | | | | | | R | X | | | | X | X | | | X | | | |
| [26] | X | | | | | X | | | | | | | R | X | X | | X | X | X | | | X | | | |
| [40] | X | | | | | | (X) | | | | | | R | X | | | X | X | | X | | | | | |
| [41] | X | | | | X | | | X | X | X | | | R | X | | | X | X | | | X | | X | | |
| [42] | X | | | | X | | | X | (X) | | X | X | R | X | | | X | X | | | X | X | X | | |
| [34] | X | | | | | | | X | X | | X | X | R | X | | | X | X | | | X | | | | |
| [33] | X | | | | | | | X | X | | | X | R | X | X | X | X | X | | | | X | | | |
| [25] | | X | | | | | | X | X | X | | | S | X | X | X | X | X | | | X | | X | | |
| [43] | X | | | | | | | X | X | | X | | R | X | X | X | | X | | | (X) | X | X | | |
| [44] | X | | | | | | | (X) | X | | (X) | | R | X | X | X | X | X | | | X | X | X | | |
| [45] | X | | | | | X | | | (X) | | | | R/S | X | | | | | X | | X | X | | | |
| [46] | | | | | | X | | | | (X) | (X) | | R | X | | | | X | | | X | X | | | |

DM, DS: digital model, shadow, FL: Fuzzy logic, GA: genetic algorithms, CA: context awareness, FLM: full lifecycle management, Auto: autonomy, CE: continuous evolving, ED, PD, HD: environmental, product, and handler data, R: real, S: simulated, IS: information system.

Forecasting tasks have been addressed with a wider range of ML techniques, with NNs still being the most widely used. Ref. [35] used a neuro-fuzzy model to forecast the future arrivals of stock keeping units (SKUs), while [40] used backpropagation NNs for sales predictions. Gradient-boosting decision tree (GDBT) has been used for anomaly detection and maintenance monitoring, as demonstrated by [36], and proximal policy optimization (PPO) has been used for inventory predictions by [38]. Ref. [41] used a combination of time-weighted linear regression method (TWMLR) and non-dominated sorting genetic algorithm (NSGA-II) for process time prediction and optimal allocation of trolleys for material handling tasks. A DT framework can curate multiple algorithms and subject problems to a set of optimization tools. Numerous ML techniques have been implemented in the "algorithm center" by [42], where the appropriate algorithm is selected to match the problem.

Supervised learning has been the most used ML method, with 11 papers utilizing it on its own or in combination with other types of learning (cf. Figure 5). Unsupervised learning has been used three times, while reinforcement learning has been used only once. Deep learning techniques, particularly deep neural networks, have received significant attention due to their increased computational power and compatibility with big data [26,30,31,38,43]. Some algorithms have been approved through testing and are envisioned to be used in a DT framework. In contrast, others have already been applied in a case study demonstrating a united DT/AI embedded system. The natural and strong relation between AI and DT could also be noticed in the bibliometric networks that showcase keywords such as "Artificial Intelligence", "Machine Learning", "Deep learning" and "Big Data". However, they were not explicitly targeted in both Figures 2 and 3.

### 5.1.2. Data

With the exception of [25,35,38,45], all DTs and AI/ML algorithms were trained using, or claimed to be connected to, mainly real product data. These studies highlighted the risks of using simulated data, which could result in unexpected outcomes. For instance, RL algorithms may exploit virtual models to terminate inventory ordering closer to the end of the simulation, minimizing holding costs.

Handler data are primarily used for online location tracking [33,35,43]. Operator-related data in general has been scarce and far removed from reality when modeling a DT of manual activities. This is probably because advances in integrating human factors into DT and simulation remain limited [47]. Ref. [48] discussed the integration of digital twins with industry 4.0, specifically from a human perspective. The paper highlights the lack of available solutions that address the human factor within DTs and proposes an approach that involves the use of data science and AI classification techniques in the form of a human skills modeling engine and a human scheduling engine to enhance digital twins of semi-automatic production lines by digitizing operator skills.

Alternatively, environmental data, although heavily talked about in the literature, are not subject to much application and have yet to be widely applied. Temperature, as an environmental factor, plays an important role in safety monitoring [31,34], and proper temperature monitoring and environmental control are critical for ensuring the safety and quality of food products in the food industry [49]. Ref. [27] suggests that all data types can be exploited for object identification through AI. DTs can develop auditory and visual signatures based on all kinds of data collected to identify changes in an environment and act accordingly.

One common goal among all DT adaptations in the literature is to access information in real-time either through IoT or IS. Even if not applied in the specific studies, there is a consensus that WDTs must be connected to the IoT and IS primarily used to describe and manage the physical twin. IS, primarily through ERP or WMS, helps manage and optimize warehouse operations. It typically includes functionalities such as inventory management, order fulfillment, receiving and shipping of goods, and tracking of warehouse activities. IoT is widely used in the literature to access and condition resources, whether they are

human or material. Ref. [46] demonstrated how combining digital twin technology, RFID technology, and spatial-operational multivariate simulation can optimize internal transport in a non-stacking warehouse. However, real-time monitoring is not always necessary, depending on the level of abstraction and the objective. More realistic, periodic, and synchronized data updates could be considered, which could be efficient, effective, and consume less energy and calculations for their applications. In other words, the level of detail and frequency of data updates should be tailored to the specific application and objective of the DT.

### 5.2. How Is AI Employed to Ensure and Elevate WDT Characteristics?

There is no standard mold for a DT. The concept of the DT has evolved over time, and until recently no standard definition existed. However, ref. [18] has provided valuable insights that have clarified the scope and potential of DTs. Despite these efforts, some studies continue to use the DT term interchangeably with simulation or cyber-physical systems, which can be misleading. AI algorithms have also become integral to digital twin models and frameworks. Nearly all examined papers developed models and architectures that integrated ML with the digital twin platform or cloud or discussed the importance of doing so. These algorithms are also tightly coupled with the virtual twin, as they could be used to enable DT characteristics to obtain more realistic DT models.

### 5.2.1. Context Awareness

AI has been successfully applied in complex configurations to generate contextual information that DTs should act on. When it comes to classification, image classification algorithms could be employed to evaluate the quality of bananas, allowing the DT to determine the appropriate actions to take, such as preservation, sale, or donation, before the fruit becomes unsuitable for consumption [30]. Similarly, ref. [35] employed machine learning models to forecast order arrivals within the hour, enabling the system to optimize the inbound synchronization strategy.

Reinforcement learning algorithms are naturally attuned to the context in which they operate, as they learn from the set of rules and interactions within their environment. Ref. [41] leveraged AI to predict the remaining processing times and compare them to the current status, thereby proactively optimizing the material handling method. Furthermore, neural networks were used to detect abnormal stationary states that may pose safety hazards in warehouse operations, helping to distinguish genuine alarms from false ones.

These examples illustrate the power of AI in generating actionable insights from complex data and enabling DTs to make informed decisions in real-time. By harnessing AI's potential, DTs can streamline operations and improve warehouse management's overall efficiency and quality.

### 5.2.2. Autonomy

Despite the immense progress made in the field of DTs and AI, showcased through the literature review, it remains an undeniable fact that human involvement is still deemed essential when it comes to the modification of the physical system or the update of the digital twin. The importance of human intervention is particularly emphasized in ensuring safety and preventing potential risks that may occur in the system in the case of a connection crash of the presence of disturbing events. Nevertheless, the interconnectivity of DTs with other software in the cloud and the Internet of Things (IoT) to the real world has given rise to more autonomous control and data acquisition. As such, the use of AI has significantly improved the efficiency of DTs in decision-making and provided them with the capability of finding optimal solutions that may surpass human perception. Nevertheless, there is still a lack of research on implementing these solutions in real-life scenarios.

To address this issue, some researchers have proposed using an architecture that involves multiple autonomous, interconnected sub-systems within the DT. This approach allows for more efficient access to data, enabling the DT to make more informed and timely

decisions. For instance, the interconnectivity of DTs with information systems to improve their autonomy has been explored in the literature [37,41,49]. However, it is important to note that even with these advancements, human presence and intervention remain a crucial component of the DT's decision-making process.

5.2.3. Continuous Evolving

The ever-changing nature of warehouses demands that DTs be adaptable and capable of accurately representing the environment, regardless of the degree of abstraction. However, this aspect has not been explored in depth in the literature. Adaptive DTs must incorporate collected data and reintroduce it into the cloud to reevaluate the virtual counterpart. Ref. [36] proposed a system with two data streams, one for real-time processing and deployment of the ML algorithm at the edge layer and the second for preprocessing, feature extraction, and updating the model in the cloud. This approach enables the DT to stay aware of contextual changes in the warehouse. Ref. [31] developed an online self-adapting mechanism to ensure that the model aligns with the environmental changes of the warehouse. In contrast, ref. [33] used a closed-loop structure that continuously updates the datasets and regenerates the programs following the gene structure, resulting in self-conscious and self-modifying algorithms.

5.2.4. Full Lifecycle Management

It is important to consider the entire lifecycle of warehouses when implementing DTs, as most studies in the literature focus solely on the middle of life. However, the RECLAIM project, presented by [28], proposes a framework that covers the entire equipment lifecycle. The goal is to extend the life of the equipment using DTs and by collecting and analyzing data, prescribe refurbishment and remanufacturing actions on the machines to restore their functionality to an "as-new" state and optimize EOL outcomes. This approach not only prolongs the useful life of the equipment but also has environmental and economic benefits. By minimizing the need for new equipment, resources and energy are preserved, and costs are reduced. Furthermore, EOL outcomes are optimized, reducing waste and promoting sustainability. Therefore, focusing on the entire lifecycle of warehouses in implementing DTs is crucial for sustainable and efficient operations.

## 6. Discussion and Further Research Perspectives

ML technics have been mainly utilized in this literature review to fulfill an application within the DT. A few articles that have not used AI explicitly still consider it an important part of their future studies [20,28,44].

AI can help DTs reach maturity and wisdom throughout the entire product lifecycle by playing two roles [50]:

- Reconstruction: AI can be an important tool for the reconstruction process, creating and revisiting the virtual representation based on the raw data from the sensors.
- Application: Once the digital twin is reconstructed, another AI algorithm can be applied to the semantically rich representation of the digital twin to support the business goals.

The literature about WDTs is lacking in research regarding the modeling of intralogistics processes and facilities in their entirety. None of the papers reviewed focused on optimizing package preparation, which includes packaging, accumulation, sorting, and shipping. This research gap warrants attention, particularly in co-packing, prospective package preparations, quality testing of packages based on client feedback, and establishing links between packaging and shipping methods as these are problems that could be recurrent and tedious.

While digital twins (DTs) are often described as the exact replica of the physical system, able to copy and anticipate every change, such a goal is not easily attainable. Full real-time connectivity and capturing every state and minimal system change is not yet feasible. The idea presumes the existence of sensors everywhere to capture every shift in the air,

ignoring the limitations of connectivity and simulation runtimes. The digital twin paradigm might be based on concrete concepts such as simulation and IoT but the twinning factors necessitates innovative and sophisticated model structures and frameworks, particularly in situations where data are scarce or unreliable, or if human involvement is present, which is usually the case in warehousing. Another field that has considered using digital twins is the medical field, particularly for the management of emergency departments which are known for being stressful and volatile environments prone to unpredictable changes [51,52]. Ref. [52] proposed a digital twin that can run multiple simulations to predict potential outcomes of a scenario. The digital twin model is updated with each synchronization of the physical system through the IS but it relies heavily on human cooperation and work ethic. Furthermore the decision variables in their model are mostly static and can be changed manually or between experiments, but they will remain constant during a simulation run. Insufficient understanding of the interplay between humans and digital twins within a work system may lead to substantial costs, misallocation of resources, unrealistic expectations from DTs, and strategic misalignments [17,53,54]. However, it is possible to envision the design and management of a stochastic WDT effectively with a certain level of abstraction and operational synchronization. The utilization of evolving, dynamic, and traceable models is of the essence in order to accurately represent and predict the behavior of DTs in a constantly changing environment [17,55]. It is even believed that the significant discrepancy between simulation results and actual system behavior is primarily attributed to the inaccuracies in the model and its parameters [55]. It is crucial to recognize and trace down the existence of unpredictable events, such as degradation of production components and unusual disruptions in physical processes that could result in an inconsistency between the digital twin model and the actual performance of its physical counterpart. This however further solidifies the role of AI in DTs as they must also be context-aware and evolve continuously to reflect these changes. AI algorithms are utilized in the development of contextual information for decision-making in warehouse environments, which are constantly changing. By analyzing vast amounts of data, AI algorithms understand and model the behavior of the system, leading to the generation of various case specific scenarios for the decision-making process. In rapidly changing environments such as warehouses, it is imperative that the digital twin's structure permits continuous adaptation to avoid false predictions and incorrect assumptions, a point that has yet to be thoroughly explored in the literature.

Keeping realistic expectations is crucial when discussing DTs and AI. Some ML algorithms are mostly black box models that we do not fully comprehend, making the technique untrustworthy. However, it can help us discover patterns we did not initially consider. Data-centric engineering, a field that leverages the best of physics, simulation, and data science, has been discussed by [20], which helps to ground AI and make it more predictable.

Another potential research gap is continuous learning (CL) in a DT framework. CL represents the model's ability to continue evolving and regenerating from a data stream, achieving a degree of autonomy without requiring human intervention. However, the concept of autonomy is poorly covered in scientific literature. Autonomous systems pose more safety hazards in the workshop as we lose control over the actions and their timing.

Another angle that is still uncharted when discussion both DTs and AI together is full lifecycle management. DTs have been associated with product lifecycle management since the very beginning. The first time the notion was ever explicitly used was by Grieves during a PLM lecture [18]. However, none of the papers from this literature review covered the entire lifecycle from a general perspective or focused on the end of life of a warehouse. None of the applications use artificial intelligence or simulation to model the beginning of the life of a warehouse, revamping and redesigning the building, or discuss what would become of the warehouse by the end of its life. IoT aside, warehouse operations can easily reach many records and data used for management or assurance purposes. Ref. [56] used machine learning models to predict warehouse component design based on data and metrics collected through the life of another storage system. The algorithms could be used

to assess the effectiveness of the current, up-and-running warehouse to either duplicate or avoid making similar mistakes. This brings us back to the ultimate research question of all time: What came first, the digital twin or the physical system?

If we consider digital twins as a simulation-based concept, then the making of the digital twin does not necessarily need the physical counterpart. It is irrelevant whether the real counterpart already exists in the physical world or is about to be constructed. Simulation will allow to test if the soon-to-be warehouse can handle different parameters such as the envisioned inventory, if resources are enough for ramp-up, test if we can get away with a traditional manual warehouse or if we ought to invest in a mechanical one ... All the previous attributes are to be consolidated and evaluated, if not changed as both twins grow and evolve. Figure 6 showcases the warehouse twins' evolution through time. This model was inspired by Sacks et al.'s (2020) representation of the lifecycle of twins for building construction. The model ensures information saving and visibility in a structured and evolutive configuration. Our vision for WDT will chronologically follow the natural evolution detailed in Figure 6, from a digital model to a digital shadow, and lastly a twin. The concept is to be constructed based on simulation software allowing us to control the environment and run experiments in the first place, import datasets, and use machine learning-based optimization and forecasts, depending on the problem. In addition, depending on the problem, we need to define the degree of synchronization required for the optimization to make sense. This way, the "digital shadow" will be equipped with automatic data exchange and become a twin.

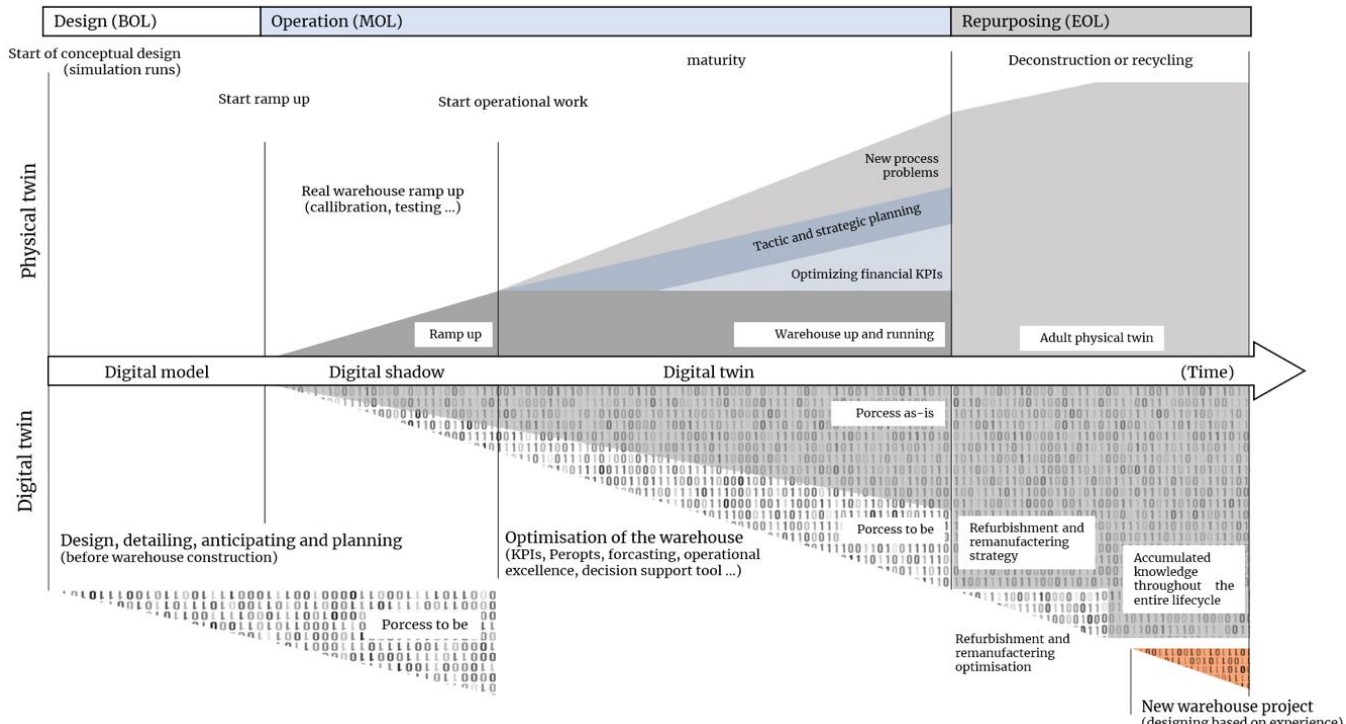

**Figure 6.** Full lifecycle management of WDT.

A few surprising studies about DTs of the supply chain (SC) made their way into the search query, mainly because of the use of the keyword "inventory" which was more managed on the scale of the supply chain management (SCM) and sales department as a whole to minimize the bullwhip effect and optimize network design performance [49,57]. We predict that the evolution of both SC digital twins and WDTs will eventually converge into one network similar to a system of systems. Research on network design, supply chain risk, and visibility still lacks adequate knowledge and precise information on each node. The mass digital transformation, currently in its infancy, displays promise of a research gap in the design and utilization of supply chain digital twins in which every node is in itself a

digital twin of a facility allowing for a fully connected, highly adaptive, interactive, global, and proactive SCM tool.

This literature review serves as a cornerstone, fortifying the foundation of our research project. Our endeavor focuses on the development of digital twins for eco-responsible intralogistics processes, fostering collaboration between the esteemed entities: Square Management Consulting Group, Square Research Center, Arts et Métiers Paristech, and Polytechnique Montréal. Furthermore, we gratefully acknowledge the financial support provided by the Association Nationale de la Recherche et de la Technologie (ANRT). Through this academic and rigorous exploration of digital twins, we have successfully identified essential training requirements and criteria, enabling us to create models that remain faithful to their real-world counterparts while simultaneously pushing the boundaries of innovation. The application of artificial intelligence has emerged as a pivotal tool, not only for practical decision-making applications but also for construction of the DT and ensuring its characteristics, an uncharted territory in the existing literature. Future work shall be centered upon guiding industrial establishments in their journey towards the advancement of virtual replicas, deftly addressing the challenges that may ensue within warehouse environments, including but not limited to data accessibility, process intricacies, and the integration of human labor.

## 7. Conclusions

The digital transition plays a crucial role in realizing the objectives of industry 5.0, enabling businesses to make informed decisions using data analysis and analytics that promote sustainability, social responsibility, and resilience. Digitalization also facilitates the creation of more efficient and interconnected systems that can adapt to changing circumstances and ensure business continuity. This literature review challenges the existing literature on digital twins, one of the industry 4.0 technologies, about its links and synergies. A systematic literature review methodology was employed to select and analyze 22 research articles, utilizing an analytical framework composed of four axes: DT requirements, intralogistics activities, data, and AI. First, the method elements were reviewed to analyze the techniques and tools required for designing an AI-embedded DT. Second, the data categories and sources used for DT modeling were identified and evaluated. Third, a use-case analysis recognized the applications of WDTs. Finally, the paper highlighted the growing potential of AI and DT to optimize warehouse creation, management, and transformation. AI algorithms have been leveraged in various ways to ensure WDT application and achieve business objectives. Furthermore, AI has shown great potential in providing DT characteristics through data analytics, reinforcing that DTs are inherently "smart" and that AI is one of its indispensable pillars. Conversely, DT presents a realistic, sophisticated, and interactive modeling environment for AI applications, providing accurate representations and data for testing and training when such information cannot be obtained from the real world. However, research on intralogistics as a whole and certain in-store activities, such as preparation and delivery, is lacking in the literature on WDTs.

**Author Contributions:** Conceptualization, A.D.E. and E.T.V.; methodology A.D.E., R.P. and E.T.V.; investigation, A.D.E.; writing—original draft preparation, A.D.E.; writing—review and editing, M.-J.B., S.L., A.A.E.C., R.P. and E.T.V.; supervision, S.L., A.A.E.C., R.P. and E.T.V. All authors have read and agreed to the published version of the manuscript.

**Funding:** This research received no external funding.

**Institutional Review Board Statement:** Not applicable.

**Informed Consent Statement:** Not applicable.

**Data Availability Statement:** No new data were created or analyzed in this study. Data sharing is not applicable to this article.

**Conflicts of Interest:** The authors declare no conflict of interest.

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
