# Peer review of "The Role of AI in Warehouse Digital Twins: Literature Review†"

_applsci, doi:10.3390/app13116746_

Round 1

Reviewer 1 Report

Digital twinning is one of the enabling technologies for implementing the Industry 4.0 paradigm.  The article aims to present a systematic literature review of the perspectives of applying  AI for warehouse digital twins. 

General concept comments
Article: The Authors, in a well-documented way, describe applied keywords for bibliometrics text mining, and present the fundamental concepts of selected literature sources to effectively presents the potential role of AI as enabling technology of transition digital twin of logistics systems to Industry 5.0 era.

Review: The article is well structured with documented and commented performed literature review. It is a valuable source of information for identifying the scope of future research on implementing AI in digital twins of warehousing processes.

Specific comments

1.      Consider improving the Keywords using upper letters ( Digital Twin, …).

2.      Please, follow editorial requirements on text formatting for enumerations in the text and bibliography.

3.      Please, include source indication for figures and tables.

Author Response

Thank you for your positive feedback on the article. We appreciate your comments and suggestions for improvement. Below is a positive response addressing each of your specific comments:

  1. We agree that using consistent capitalization for keywords is important for clarity and consistency. the keywords have been updated accordingly to enhance readability.

  2.  We apologize for any inconsistencies in the formatting and we have made sure to align with the required guidelines in the final version of the article.

  3. We ensured that each figure and table in the article includes clear source indications in the final version.

Once again, we would like to express our gratitude for your valuable feedback. If you have any further suggestions or questions, please feel free to let us know.

Reviewer 2 Report

The authors have written a very interesting article, the subject of which is extremely up-to-date. They meticulously presented the scientific methodology and conducted a selection of research literature. Undoubtedly, AI will be used in many areas in the near future, and the selected warehouse area is one of them.

Conducting a scientific argument, analyzing the collected data, and finally developing research conclusions or even their graphical presentation proves the scientific maturity of the authors of this article.

I really enjoyed reading and reviewing this article.

Congratulations to the Authors!

Author Response

Thank you for your valuable feedback. The authors and I, Adnane, greatly appreciate your positive comments. Your review has brought us immense joy and serves as a testament to the effort and dedication we put into this article.

Best regards.

Reviewer 3 Report

The paper presents a systematic literature review on the role of AI in digital twins for warehouse management. The paper proposes an interesting contribution, given the level of novelty of the topic, but there are some minor refinements to be implemented before publication.

1) The paper begins by citing the industry 5.0 paradigm. Nevertheless, the use of AI and DT already belongs to the industry 4.0 paradigm. If industry 5.0 is to be introduced this must be well justified. 

2) The preceding error may stem from the authors' incorrect definition of industry 5.0, namely "industry 4.0 [...] in which the physical and digital worlds are increasingly intertwined." Otherwise, Industry 5.0 which is nothing but industry 4.0 with focus on preservation of resources, climate change and social stability. Consequently, the introduction and use of these two terms need to be corrected. Note that "merely" speaking of Industry 4.0 is not improper....

3) The introduction starts with rather trivial or outdated considerations (the phrase "Material handling, in particular, can account for 15% to 70% of production costs" attributed here to [Glatt, M.; Sinnwell, C.; Yi, L.; Donohoe, S.; Ravani, B.; Aurich, J.C. Modeling and Implementation of a Digital Twin of Material 729 Flows Based on Physics Simulation. Journal of Manufacturing Systems 2021, 58, 231-245] is actually the old and well-known phrase by Tompkins in his famous 2009 book "facilities planning" (!!! The authors can easily trace it in the bibliography of the authors they cite). Anyway: after an obvious introduction, the topic of AI in DT for warehouses is introduced very quickly. For proper framing of the topic and ease of reading, the value brought by simulation in warehouse planning and management should first be introduced, thus arriving at the DTs of Industry 4.0. A possible reference in this regard is, in the same journal, the paper by D'Orazio et al. (2020) Industry 4.0 and World Class Manufacturing Integration: 100 Technologies for a WCM-I4.0 Matrix, Appl. Sci. 2020, 10, 4942; doi:10.3390/app10144942. In this sense, the introduction of the paper needs to be radically revised and the first part, framing, improved.

4) The English needs to be improved and there are some formatting typos (blanks, dots, etc.).

Author Response

Thank you for your valuable feedback. The authors and I, Adnane, greatly appreciate your positive comments. We have taken the time to correct our manuscript following the suggested elements, namely:

  1. The introduction to industry 5.0 has been revised to highlight its societal engagements, particularly sustainability, social responsibility, human factors, and resilience in the development of digital twins. We wholeheartedly agree that both AI and DT belong to the industry 4.0 paradigm, and we have accordingly corrected the article. "Digital twin (DT) is one of the pillars of industry 4.0..."

  2. We agree with your point, as we specifically mention that "With technology as the enabling tool and societal needs as the goal, industry 5.0 aims to promote sustainability, social responsibility, and resilience by integrating digital technologies." We have also corrected the article by adding references [5,6] that provide the definition of industry 5.0 and support the reviewer's point. We have also included a reference from this journal that discusses the ethical implications of industry 5.0. The introduction of Industry 5.0 primarily indicates the new context and societal objectives that companies need to uphold. Throughout the article, we adhere to the technological requirements and terminology of industry 4.0 as the reference.

  3. Thank you so much for these insightful comments. We have reevaluated the structure of the introduction to incorporate the proposed improvements. In this manner, we first introduce the context, followed by how DT is the natural extension of simulation, which has proven its merits as an industry 4.0 technology improving manufacturing and logistics (we have used the reference you kindly provided). These applications can be further enhanced with the adaptation of DT as "connected" simulation modules. Then we present how data has become more available in warehouses and warehousing activities but is often underutilized, offering significant potential for AI and DTs as data-hungry technologies.

  4. We have made sure to correct the article to ensure that the English level is up to par.

Please let us know if you have any further suggestions or if there are any other areas we should address. We appreciate your input and are committed to improving the quality of our manuscript.

Reviewer 4 Report

Overall, I consider the contribution to be good in terms of the literature review in the discussed field. However, I feel that it lacks a significant original scientific contribution and a proposed methodology by the authors that could be applied to the development of artificial intelligence, digital twins, and their interaction in the field of warehousing. I appreciate the conducted analyses and their evaluation. I would recommend the authors provide more descriptions and highlight their own proposals and the objectives derived from them in processing an extensive literature review on the given topic.

Author Response

We would like to thank you for your review and constructive feedback on our article. We appreciate your positive assessment of the literature review conducted in the field of AI and digital twins in warehousing. We understand your suggestion regarding the need for a significant original scientific contribution and a proposed methodology by the authors.

We have taken your comment into consideration and have made significant improvements to the article to address this concern. In particular, we have added a paragraph at the end of the discussion that puts the positioning of our project and future work in better light.

By incorporating this paragraph, we aim to provide a more comprehensive overview of our project's originality and scientific contribution, for the development of artificial intelligence and digital twins in the field of warehousing as this is only the first step form me, Adnane, as a first year phd candidate. We hope that this addition addresses your suggestion and adds value to the article.

Once again, we sincerely appreciate your feedback, as it has helped us enhance the quality and relevance of our work. Should you have any further suggestions or questions, please do not hesitate to let us know.